# Gossypiboma, the Hidden Enemy of an Emergency Cesarean Hysterectomy—Case Report and Review of the Literature

**DOI:** 10.3390/jcm12165353

**Published:** 2023-08-17

**Authors:** Valentin Nicolae Varlas, Roxana Georgiana Bors, Bogdan Mastalier, Irina Balescu, Nicolae Bacalbasa, Monica-Mihaela Cirstoiu

**Affiliations:** 1Department of Obstetrics and Gynecology, Filantropia Clinical Hospital, 011132 Bucharest, Romania; valentin.varlas@umfcd.ro; 2Faculty of Medicine, “Carol Davila” University of Medicine and Pharmacy, 020956 Bucharest, Romania; monica.cirstoiu@umfcd.ro; 3General Surgery Clinic, Colentina Clinical Hospital, 020125 Bucharest, Romania; 4General Surgery Department, “Carol Davila” University of Medicine and Pharmacy, 050474 Bucharest, Romania; 5Department of Surgery, Ponderas Academic Hospital, 021188 Bucharest, Romania; irina_balescu206@yahoo.com; 6Department of Visceral Surgery, Fundeni Clinical Institute, 022328 Bucharest, Romania; nicolaebacalbasa@gmail.com; 7Department of Obstetrics and Gynecology, University Emergency Hospital Bucharest, 050098 Bucharest, Romania

**Keywords:** emergency cesarean section, cesarean hysterectomy, textiloma, foreign body, gossypiboma, iatrogenic

## Abstract

Gossypiboma or textiloma is a rare medical situation that can complicate the favorable evolution of a surgical case, with repercussions for the patient’s prognosis. The diagnosis can be difficult due to various clinical symptoms, the time elapsed since the surgical intervention, and the imaging often not being precise in detecting textilomas. Due to the medicolegal implications, the reporting of this event is inconsistent. We present a rare case of a 28-year-old woman who presented with vague pain in the left iliac fossa 11 months after an emergency cesarean hysterectomy was performed. The preoperative imaging examination identified the presence of a subhepatic mass with dimensions of 10 × 8 cm^2^ and another formation in the right iliac fossa with dimensions of 11 × 9 cm^2^. Exploratory laparotomy found the presence of a large subhepatic gossypiboma, intimately adherent to the hepatic angle of the colon and omentum and a second one adherent to the sigmoid colon, small intestine, and parietal peritoneum. The particularity of this case is given by the simultaneous presence of two textilomas with inconclusive evolution, which can make the differential diagnosis difficult to achieve. For a better assessment of the risk of occurrence of this pathology and the identification of a correct prevention strategy, we performed an extensive search and a review of all the articles published in the PubMed database, identifying 57 articles. In conclusion, emergency surgery increases the risk of this complication, and, as a result, prevention can be achieved by following existing protocols in the operating room.

## 1. Introduction

The medical terminology related to gossypiboma or textiloma defines the presence of an operating field, cotton compress, or surgical sponge left involuntarily in the peritoneal cavity during surgery [1]. The most common localization of gossypibomas is the intraperitoneal cavity [1,2]. Although the real incidence of gossypibomas is unknown, a positive evolution has been observed, marked by the decrease in detection rate in the last decades. Thus, if initially the reported incidence was 1:1000 abdominal surgeries [3], recent studies show 0.08–0.18:1000 [4,5]. The incidence regarding the identification rate of retained foreign bodies (RFB) related to the type of surgical intervention varies from 17.69% for cesarean section, 16.33% in abdominal hysterectomy, and 13.54% for exploratory laparotomy in the acute abdomen [6]. Compared to the percentage of identification of gossypibomas from the total RFB, the incidence varies between 17% and 68% [4,5]. Furthermore, Birolini et al., in a study of 4547 cases, identified a percentage of 90% of textilomas, of which the most common were large sponges [6].

The diagnosis can be early, a few days after the operation, or sometimes they can be identified after many years, and it can be associated with increased morbidity and sometimes mortality. Clinical manifestations, treatment, and prognosis of gossypibomas can vary depending on the time of diagnosis and the type and intensity of tissue reactions. Thus, early postoperative, acute, exudative, or purulent inflammatory reactions can occur, with the appearance of peritonitis and abscesses, in which the clinical signs of sepsis with fever, abdominal pain, nausea, vomiting, abdominal distention, and fatigue predominate. Late postoperatively, chronic inflammatory reactions, xanthogranulomatous, aseptic fibrosis, and calcifications can be encountered when patients can be asymptomatic or oligosymptomatic with abdominal pain, transit disorders for gases and stool, abdominal distension, or determined by the compression of the respective mass [7].

Imaging diagnosis is based on endovaginal/abdominal ultrasonography, abdominal–pelvic radiological examination, contrast-enhanced computed tomography, and magnetic resonance imaging [8]. The differential diagnosis must also be performed with other intra-abdominal tumor masses [9]. The therapeutic solution for these patients is, in most cases, surgical, this being all the more difficult the later the diagnosis is established. Complications caused by perforations, fistulizations, and obstructions require the intervention of a general surgeon in the surgical team due to intraoperative technical difficulties.

Although we accept that there is a risk of this undesirable medical situation occurring during surgical interventions due to the medicolegal implications, most surgeons (74%) mentioned that they did not inform the patient about the existence of a foreign body, invoking other possible causes of the indication for re-intervention [6].

This article aims to evaluate the magnitude and risk of gossypibomas after cesarean sections, especially emergency ones, to identify the risk factors and reduce their incidence.

## 2. Case Report

The 28-year-old patient presented to the Filantropia Obstetrics-Gynecology Clinical Hospital, Bucharest, with chronic lower abdominal pain, especially in the left iliac fossa. The patient had two cesarean sections in another hospital (in 2018 and 2021, respectively). The last one was performed 16 months ago for uterine rupture prophylaxis, followed by an emergency supracervical hysterectomy due to the formation of a pelvic hematoma.

The clinical symptoms were dominated by chronic pain in the left iliac fossa, which progressively increased in intensity. A well-defined tumor mass was palpated in the left lower abdomen during clinical examination. The bimanual pelvic examination revealed a tumor mass with dimensions of 11 × 9 cm^2^, increased consistency, and reduced mobility, which was slightly painful. Blood count, liver, and kidney function tests were within normal limits. The ultrasound examination (Figure 1a–c) reveals a surgically excluded uterus with a 2.9 cm long cervix, 3.5 × 2.8 cm^2^ right ovary; the left ovary is not visible, instead a complex mass is identified in the left lower quadrant (left iliac fossa) with dimensions of 9.6 × 5.4 × 4.8 cm^3^, with hyperechoic areas alternating with anechoic areas, without highlighting a capsule, without Doppler signal. A mass with the same characteristics with dimensions of 8.5 × 4.9 cm^2^ was identified in the upper right quadrant. The MRI (Figure 1d,e) reveals expansive formations, ovarian on the left side and paracaval in the mid-paramedian abdominal floor on the right side, with an appearance suggestive of endometriosis cysts.

Elective exploratory laparotomy identified the presence of the two encapsulated masses adherent to the neighboring structures. During the inspection of the peritoneal cavity, adhesions of the omentum at the level of the anterior parietal wall are observed, as well as a surgically absent uterus, the appendix with right fallopian tube, and the ovary adhered to the level of the ascending colon. In the left iliac fossa, a tumor mass of approximately 12 × 10 cm^2^ is present, to which the parietal peritoneum, descending colon, sigmoid colon, and intestinal loops are adherent. It was completely removed en bloc with the omentum, left ovary, and fallopian tube. The second tumor mass of approximately 10 × 9 cm^2^ in the upper right abdominal quadrant was intimately adherent to the anterior parietal, peritoneum, the ileal loops, and the hepatic angle of the colon.

The sectioning of the first formation with dimensions of 9.5 × 9.5 × 6.5 cm^3^ identified a gauze mesh (Figure 2), which was resected with a portion of the omentum. The postoperative evolution was without complications, with the patient being discharged on the fifth postoperative day.

The macroscopic examination revealed:-Left adnexal tumor mass measuring 9.5 × 9.5 × 6.5 cm^3^; when sectioning, textile material with dimensions of 9 × 8 × 5 cm^3^ is evacuated; after the extraction of the textile material, the internal surface of the pseudocyst wall is intensely congestive, with greyish-yellow deposits; and isolated intramural nodular mass with dimensions of 4.5 × 4.5 × 1 cm^3^ of firm elastic consistency;-Subhepatic tumor mass with dimensions of 10 × 8 × 5.5 cm^3^ with a grayish-pink external surface with areas of fatty tissue; when sectioning, textile material with dimensions of 7 × 6 × 4 cm^3^ is extracted.

The microscopic examination highlighted:-Left adnexal tumor mass: tissue fragment with a histopathological aspect of conjunctive-adipose and vascular–nervous tissue presenting multiple foci of chronic granulomatous inflammation with multinucleated foreign body giant cells arranged around exogenous, acellular, translucent materials; diffuse areas of fibroblast–fibrocystic proliferation; numerous groups of foamy histiocytes, some with a multinucleolate appearance; and marked capillary hyperemia, interstitial edema, and diffuse regions of hematic extravasation. The ovarian histological structure is identified at a certain level, with multiple foci of chronic granulomatous inflammation with foreign body multinucleated giant cells, in addition to tubal wall with lesions of chronic xanthogranulomatous salpingitis, discrete tubular epithelial hyperplasia, moderate capillary hyperemia, and intramural interstitial edema;-Right subhepatic tumor mass: tissue fragment with a histopathological aspect of conjunctive-adipose and vascular–nervous tissue presenting multiple foci of chronic granulomatous inflammation with multinucleated foreign body giant cells arranged around exogenous, acellular, translucent materials; diffuse areas of fibroblast–fibrocystic proliferation; numerous groups of foamy histiocytes, some with a multinucleolate appearance; and moderate capillary hyperemia, interstitial edema (Figure 3).

## 3. Discussion

Gossypiboma or textiloma represents an important medical event due to the medicolegal implications and an increased risk of morbidity and mortality [1]. Furthermore, the actual reporting of these cases is inconstant, with the global incidence reported in abdominal surgeries being between 1 and 1.2 per 1000–1500 [9]. Another study carried out on 49,831 surgeries under general anesthesia identified 24 cases of retained foreign bodies (0.48:1000), of which 17% (4 cases—0.08:1000) were gossypiboma [5]. It has been observed that the increased incidence is associated with emergency surgical interventions, especially those in the obstetric field (placenta previa, placenta accreta, hemorrhage, uterine rupture). Emergency surgical interventions represent the most frequently encountered risk factor regarding the appearance of gossypibomas (26%), followed by wrong counting of sponges (25%) [6]. The mechanisms underlying the increase in risk are non-compliance with operating protocols, lack of coordination of the surgical team, lack of training of the medical staff, modification of the initial operating plan through the participation of a multidisciplinary team, improper counting of the textile material due to increased blood loss, long operations and laborious, intraoperative instability of the patient, increased BMI, and comorbidities [9].

We performed a comprehensive electronic search in the PubMed database, the search was from inception to 31 May 2023, where we identified 57 published cases with the MeSH search terms “gossypiboma”, “textiloma”, “gauze”, “sponge”, and “cesarean section” (Table 1).

The average duration from cesarean section to the time of diagnosis of gossypibomas was, on average, (±SD) 3.69 ± 6.24 years (range from 0.04 to 29 years), which indicates that in most cases, the clinical manifestations have been asymptomatic or oligosymptomatic. This is also highlighted in the study by Birolini et al., in which asymptomatic and oligosymptomatic clinical manifestations represented 12% and 71% of cases, respectively [6]. The average age of the patients from all the studied cases at the time of diagnosis was 34.58 ± 8.38 (range from 20 to 68 years), representing an independent factor of the incidence of gossypibomas.

Although a series of studies showed an increased rate regarding the time of diagnosis of gossypiboma being within the first two months, the analysis of the articles studied in this review highlighted a rate of 19.29%, with a peak of 47.36% at more than a year after the surgery. This fact is possibly correlated with the intensity of the clinical manifestations, the severe ones being found in only 17% of cases [6]. The body’s response to the intra-abdominal presence of textile material, depending on the time elapsed until the diagnosis is established, is based on the appearance of an aseptic process of a fibrinous nature or a local exudative process that allows the formation of an abscess [7].

The clinical manifestations can be varied and atypical, depending on the topography, the size of the textiloma, their number, and the possible complications that may occur, such as subacute intestinal obstruction and peritonitis. In the early detection of gossypiboma, pain is the main symptom, with palpations of abdominal formations, or it can be asymptomatic/oligosymptomatic, being detected later, after a few years [6,43]. In the presented case, the gossypiboma was diagnosed 16 months after the emergency cesarean supracervical hysterectomy due to chronic pain in the lower abdomen and the presence of a tumor mass that deformed the abdominal wall.

In the natural evolution of foreign objects retained in the abdominal cavity, processes of encapsulation through fibrotic reaction, intraluminal migration at intestinal, vaginal, and urinary bladder levels, and the formation of abscesses or fistulas can be encountered [2]. Thus, imaging to detect gossypiboma is limited, potentially leading to diagnostic confusion. The differential diagnosis of gossypiboma is made with cystic or pseudocystic formations, tumor formations, abscesses, hematomas, and granulomatous formations [9].

The initial investigation is ultrasonographic, which describes the character of the formation, dimensions, structure, vascularization, and anatomical relationship with the neighboring structures. Afterward, the evaluation can be performed by using radiological investigations to diagnose an intestinal occlusive process of a modified anatomical topography and less of the formation due to the lack of radiopaque markers. Completing the imaging evaluation using CT and MRI increases the chances of preoperative diagnosis of this pathology [8,42] (Table 2).

The most frequent possible complications described secondary to the presence of gossypiboma were represented by fistulas (19.29%) [10,13,19,20,23,27,32,33,37,50], perforations (12.28%) [1,26,29,44,45,51,53], obstructions (5.26%) [11,32,37], and bladder injuries [34] (Table 3).

The therapeutic strategy of gossypiboma is surgical, being differentiated depending on the diagnosis time and possible complications’ association. Thus, the preoperative evaluation and preparation of these cases are essential, because the association of a fistula, an occlusion, a perforation, an abscess, or an extensive adhesion syndrome involving neighboring organs can be elements that complicate the surgical intervention, and prolong operative time and the duration of hospitalization. The surgical approach can be on the same incision by open, endoscopic, laparoscopic, or robotic surgery.

Late diagnosis of gossypiboma due to the intense inflammatory and granulomatous reaction with the textile material forms an important adhesive process. Early diagnosis is associated with peritonitis, requiring a quick approach to remove the textiloma. Depending on the topography of the textiloma, and the complications of its presence (fistulas, perforations), its surgical removal may involve intestinal resections and anastomoses, as well as epiploic, hepatic, gastric, and adnexal resections [1,11].

The basic prevention related to the occurrence of gossypiboma is achieved by managing the number of pieces of soft textile material and careful exploration by the surgical team of the peritoneal cavity before closing the wound. Using textile material with radiopaque thread or chips can contribute to their rapid identification [7,51] (Figure 4).

## 4. Conclusions

Although it represents a rare postoperative complication, gossypiboma is associated with a morbidity and mortality rate dependent on the initial pathology, the delay in establishing the diagnosis, and the subsequent postoperative evolution with serious ethical and medicolegal implications. The non-specific clinical manifestations, the imaging that is difficult to interpret, the unpredictable evolution burdened with complications, and the multidisciplinary approach are all challenges regarding the therapeutic management of gossypiboma. As a result, prevention is the best treatment for this pathology, achieved by following the surgical and the operating room protocols, managing the operative time appropriate to the surgical intervention’s complexity, and ensuring the training level of the medical staff. In the situation where we encounter this pathology, the surgical solution of the case and the professional deontology in relation to colleagues and patients are elements that contribute to an appropriate approach.

## Figures and Tables

**Figure 1 jcm-12-05353-f001:**
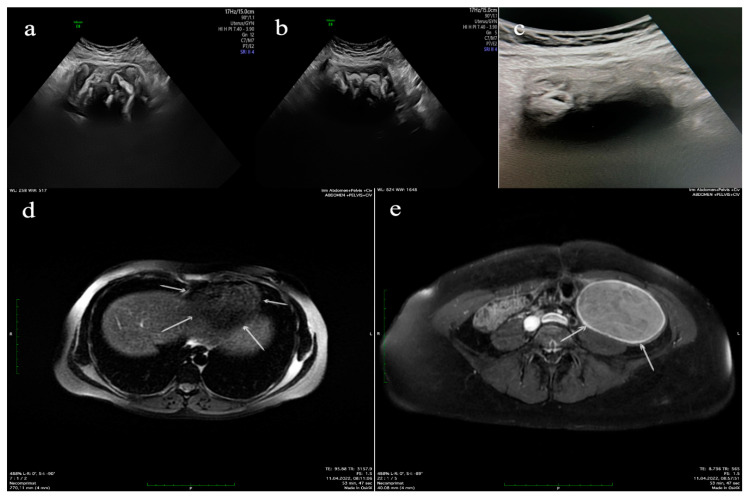
(**a**) Ultrasound longitudinal and (**b**) transversal section of a complex mass with hyperechoic alternating with anechoic areas identified in the left lower quadrant (left iliac fossa), (**c**) ultrasound scan shows an echogenic, inhomogeneous mass with dense posterior acoustic shadowing located in the right upper abdominal quadrant, subhepatic, (**d**) MRI axial T1WI section shows a round mass of hypo intensity in the right upper abdominal cavity (white arrows), (**e**) MRI T2WI-FS axial section shows heterogeneous hyperintensity with a complete hypo intensity capsule (white arrows).

**Figure 2 jcm-12-05353-f002:**
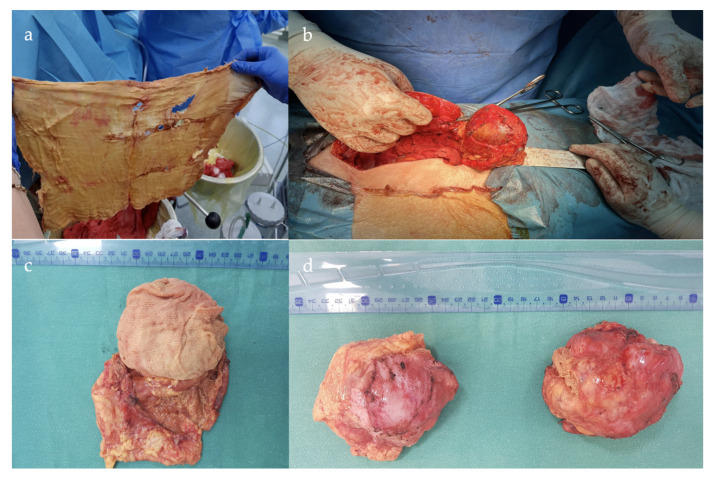
(**a**) Unfolded retained gauze towel resected from the lower left abdominal quadrant mass; (**b**) intraoperative imaging showing the gossypiboma adherent to the bowel with retained gauze seen inside; (**c**) postoperative imaging by sectioning the specimen identified a surgical gauze foreign object retained in the mass; (**d**) postoperative view of both specimens discovered intra-abdominally.

**Figure 3 jcm-12-05353-f003:**
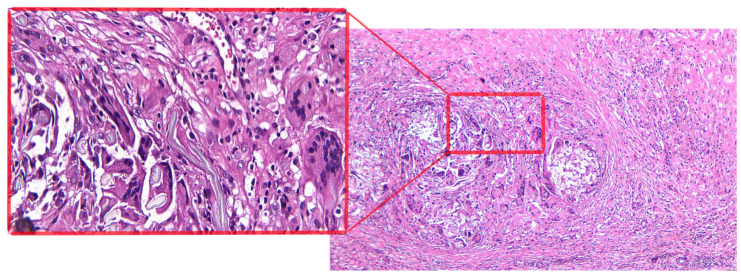
Microscopic examination showed by H&E × 100 staining a fibrous encapsulation with multinucleated foreign body giant cell reaction (in the cartridge ×400 magnification).

**Figure 4 jcm-12-05353-f004:**
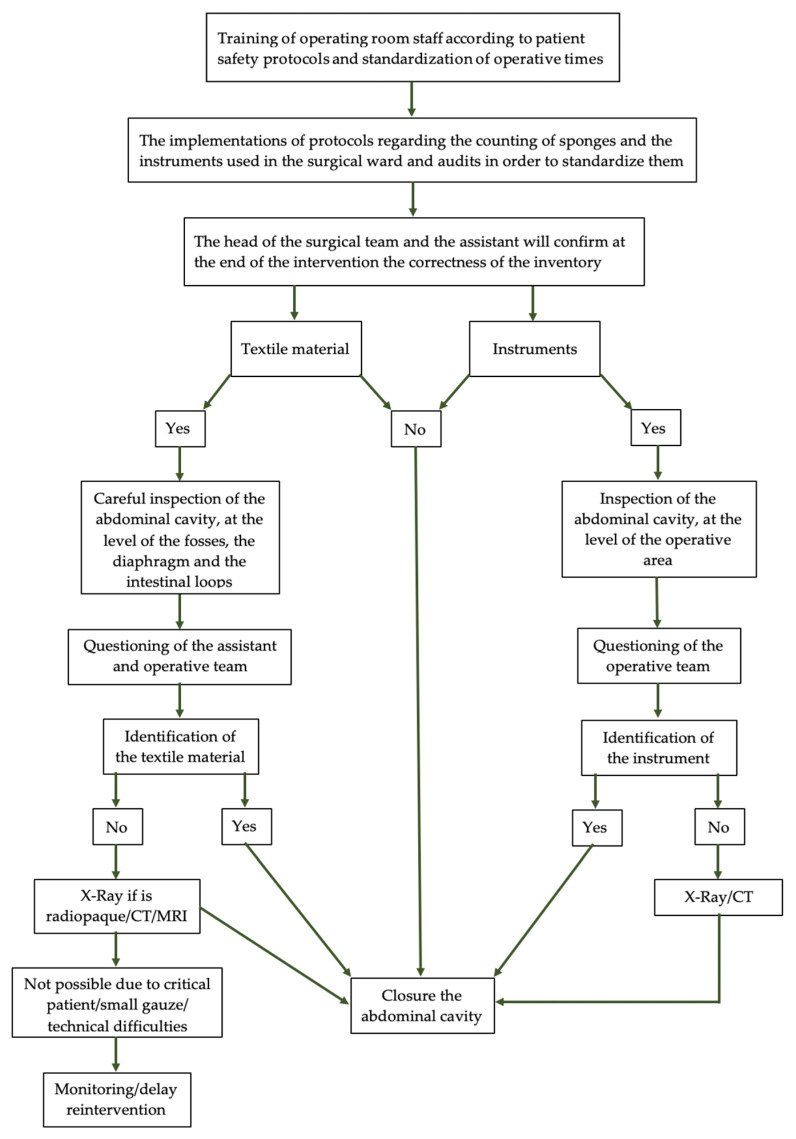
Algorithm to prevent gossypiboma/retained instrument in emergency surgical interventions in Obstetrics.

**Table 1 jcm-12-05353-t001:** Synopsis of the gossypiboma all-time search in PubMed database.

Author	Age	Obstetric Diagnosis	Type of Surgery	Clinical Manifestations	Duration since the C-Section	Diagnostic Mode	Gossypiboma Characteristics	Relaparotomy	Complications
Casal [10] 1961	N/A	N/A	CS	N/A	N/A	N/A	N/A	Resection anastomosis of the involved loops	Jejunocolonic fistula
Al-Salem [11] 1989	N/A	N/A	CS	N/A	3 years	N/A	Bowel transluminal migration of the sponge	Enterotomy and removal of the mass, resection, and anastomosis	Intestinal obstruction
Reinke [12] 1992	N/A	N/A	CS	N/A	N/A	N/A	Mid-abdomen mass	N/A	N/A
Haddad [13] 1994	27	II G, II P	CS, postpartum supracervical HT	Vaginal discharge, pelvic heaviness	6 weeks	N/A	Seven 4 × 4 cm^2^ gauzes in the vagina	Conservative management	Jejunovaginal fistula
Rajagopal [14] 2002	31	III P	CS	Abdominal mass	12 weeks	USG *, CT *	7.5 cm entangled small-bowel mass adherent to the sigmoid	Ileal resection of the fistulous segment and end-to-end anastomosis	Ileoileal fistula
Yuh-Feng [15] 2005	42	II P	CS	Incidentaloma	N/A	CT **, PET CT *	Mass in the anterior right paramedian abdomen	Yes	No
Saidi [16] 2007	40	I P	CS	Right mid-abdominal pain	1 year	USG *, CT *	10 cm mass in the right iliac fossa	Yes	No
Aminian [17] 2008	27	I P	CS	Abdominal mass	5 years	X-ray *, CT *	Retained gauze in the center of the abdomen	Yes	No
Zantvoord [18] 2008	39	I P	CS	Tiredness	12 weeks	X-ray *, CT *	Transmural migration of a 60 × 40 cm^2^ surgical gauze	-	No
Tandon [19] 2009	30	I P	CS	Colicky abdominal pain and distension	2 years	USG *, CECT *	Mass in the left lumbar region	Resection anastomosis of the involved loops	Mid-transverse colon and jejunal fistulization
Uluçay [20] 2010	22	I P	CS	Left side abdominal pain, diarrhea	7 months	Clinical *, X-ray **, USG *, CT *	15 cm in diameter mass settled in the sigmoid colon	Resection of the sigmoid colon and small bowel	Sigmoid colon and ileum fistulization
Dash [21] 2010	30	II P	CS	Abdominal pain	9 years	USG *	18 × 15 cm^2^ mass arising from the anterior and right part of the uterus	Yes	No
Patil [22] 2010	23	I P	CS	Colicky pain in left iliac fossa, vomiting	3 months	X-ray *, CT *	Mass in the lower abdomen	Proximal enterotomy to extrude the mop	No
Govarjin [23] 2010	35	II G, II P	CS	Anorexia, partial bowel obstruction	5 months	X-ray **	Retained gauze migration in the terminal ileum	Enterolysis, terminal ileotomy, right hemicolectomy with ileocolic anastomoses, and removal of the fistula tract	Periumbilical fistula
Mavrigiannaki [24] 2011	20	I P	CS	N/A	N/A	N/A	N/A	N/A	N/A
Kawamura [25] 2012	41	I P	CS	Abdominal pain	1 year	CT ** MRI **	5.5 cm diameter pelvic mass	Yes	No
Quraishi [26] 2012	24	N/A	CS	Fever, abdominal pain, vomiting	1 month	USG **, X-ray **, CT *	Retained surgical sponge in the lower abdomen	Ileal perforations followed by anastomosis	Ileal perforations
Karasaki [27] 2013	33	I P	CS	Epigastric pain, fever	6 weeks	USG *, CT *	10 cm diameter mass in the left upper quadrant	Partial resection of the descending colon, small intestine	Descending colon fistula
Rafie [28] 2013	29	I P	CS	Colicky abdominal pain and distension, nausea, vomiting, constipation	9 months	X-ray **	Jejunal transluminal migration of the sponge	Enterotomy and removal of the mass, resection, and anastomosis	No
Usta [29] 2013	30	N/A	Emergency CS	Sepsis, abdominal mass, disseminated tenderness, defense, and rebound	4 months	Clinical *, USG *, X-ray *, CECT *	10 cm diameter wide mass lesion image was seen retrovesically	Wound debridements of the abscess with uterine dehiscence and necrosis of the edges, ileal resection, and an end-to-end anastomosis	Uterine wound dehiscence and ileal injury
Kashima [30] 2014	35	II P	CS	Miction pain	11 years	Cystoscopy *	Remnant gauze migration in the bladder (a calcified mass)	Transurethral operation	No
Rehman [31] 2014	40	N/A	CS	Nausea, loss of appetite, and lower abdominal discomfort	15 years	USG **, EGDS **, colonoscopy **, CT *	Mass in the lower left abdomen	Laparoscopy	No
Srivastava [32] 2014	38	N/A	CS	Pain and a chronic lump in the right iliac fossa	4 years	USG **, CECT **, MRI **	20 × 15 cm^2^ mass in the right iliac fossa	Resection of the mass along with an area of the terminal ileum	Fistula and intestinal obstruction
Faghani [33] 2014	35	N/A	CS	Colicky abdominal pain, vomiting, and constipation	2 years	USG **, X-ray **	One sponge in the omentum and the other one in an enterocolic fistula in the distal ileum	Omentectomy and end-to-end anastomosis after resection of the fistulized segment (ileum, right colon)	Enterocolic fistula
Lee [34] 2015	38	I P	CS	Lower quadrant pain	24 years	X-ray **, CECT *	9 cm diameter mass in the right lower quadrant	Laparoscopy with bladder repair	Bladder injury
Rafat [35] 2015	22	I P	CS	Discomfort, heaviness in the lower abdomen	2 years	Clinical *, USG *	6 × 7 cm^2^ well-encapsulated mass in the pelvic cavity	Yes	No
Chopra [1] 2015	N/A	N/A	Emergency CS	Abdominal wound discharge	3 weeks	Clinical *		Abdominal wound	No
Chopra [1] 2015	N/A	N/A	Emergency CS	Adnexal mass	1 year	USG *		Yes	No
Chopra [1] 2015	N/A	N/A	Peripartum HT	Puerperal sepsis	2 weeks	USG **, CECT *	A mass in the right flank	Yes	No
Chopra [1] 2015	N/A	N/A	Peripartum HT—scar rupture	Sepsis, abdominal mass	8 months	CECT *	Buried in the lumen of the intestinal loop	Excision of the fistulous tract and end-to-end anastomosis	Gut perforation
Rabie [36] 2016	39	I P	Elective cesarean HT—placenta increta	Abdominal pain, constipation, vomiting	18 days	USG *, X-ray **, CT *	10 × 10 cm^2^ mass in the right upper quadrant	Yes	No
Rabie [36] 2016	46	II P	Cesarean HT	Abdominal pain	9 years	USG *, CT *	13 × 18 cm^2^ pelvic mass	Colonic resection	N/A
Rabie [36] 2016	35	I P	CS	Abdominal pain, nausea, fever, vomiting	2 months	X-ray **, CT *	Mass in the left lower quadrant	Yes	No
Margonis [37] 2016	36	N/A	CS	Abdominal pain, nausea, vomiting	6 months	USG **, X-ray **, CT **	The 20 × 25 cm^2^ sponge in the lumen of the small intestine.	Bowel resection en bloc with the affected sigmoid and a loop sigmoidostomy	Intestinal obstruction and ileocolic fistula
Susmallian [38] 2016	34	I P	CS	Abdominal pain, fever	9 years	CT *	Intraabdominal and pelvic mass	Yes	No
Kostandinidis [39] 2017	68	N/A	CS	Acute urinary retention	29 years	USG **, CT **	12 cm diameter mass at the left side of the pelvis	Yes	No
Oran [40] 2018	36	II P	CS	Painful mass in the left lower quadrant	15 years	USG **, CT *	11 × 9 × 7 cm^3^ mass on the left lower abdomen near the ovary	Yes	No
Kondo [41] 2018	42	II P	CS	Lower abdominal bloating	9 years	X-ray **, CT *	Two smooth masses partially adherent to the omentum and colon	Yes	No
Gavrić [7] 2018	45	II P	CS and laparotomies for retained needle	Recurrent pelvic pain	11 years	USG *	Structure with mixed echogenicity laterouterine right with a diameter of 4.9 cm	Total abdominal HT, bilateral salpingo-oophorectomy, remove gauze from the right obturator fossa	No
Fatima [42] 2019	30	I P	CS	Abdominal pain	3 months	USG *, CT *	7.2 × 4.5 cm^2^ mass in the right upper and left lower quadrant	Right hemicolectomy was done with double-barrel ileostomy	Deceased
Bilali [43] 2019	42	I P	CS	Abdominal mass	2 years	USG *, MRI *	Mass in the right quadrant	Laparoscopy	No
Mejri [44] 2020	29	I P	CS	Abdominal pain, fever	5 months	MRI *	Two collections located in the right and left iliac fossa	Sigmoidectomy with a Hartmann procedure and ileostomy	Sigmoid colon and small bowel perforation
Alemu [45] 2020	32	II P	CS	Lower abdominal pain, vomiting, nausea, transit stopped for gas and feces.	4 months	X-ray **, USG **, CECT **	In the lower left quadrant complex mass of 6 × 2.6 cm^2^ with central shadowing gas bubbles	The 10 × 8 cm^2^ surgical sponge came out through the rectum	Jejunal perforation on the antimesenteric border
Sankpal [46] 2020	40	I P	CS	Incidentaloma	5 years	N/A	15 × 10 cm^2^ mass in the gastrocolic omentum	Yes	No
Omar [47] 2020	40	I P	CS	Abdominal pain, diarrhea, bilious vomiting	4 months	USG **, CT **	Retained surgical sponge in the pelvic cavity	Resection of the involved parts of the ileum and the sigmoid colon with end–end anastomosis	Transmural erosion and ulceration of the sigmoid colon
El Zemity [48] 2020	26	I P	Elective CS	Abdominal pain, fever	18 months	Clinical *, CT *	15 × 14 × 12 cm^3^ intra-abdominal mass in the umbilical region	Yes	No
Amodeo [49] 2021	35	I P	CS	Pelvic pain	2 years	USG **	Surgical gauze in the uterine isthmus at the c-section scar site	Hysteroscopy	No
Jha [50] 2021	28	IP	CS	Abdominal pain, fecal discharge	4 months	Clinical *, USG *, MRI *	Multiple loculated abscesses, a mass of 15 × 10 cm^2^ in the left parauterine space	Colouterine fistula resection with end sigmoid colostomy	Colouterine fistula
Bairwa [51] 2021	30	N/A	CS	Abdominal pain	2 weeks	CECT *	6.2 × 6.1 cm^2^ well-defined mass in the left lumbar region	Yes	No
Bai [8] 2021	29	N/A	CS	Intermittent abdominal pain, distension, constipation	4 months	CECT *, MRI *	N/A	Enteroenterostomy	Intestinal ulcer and perforation
Bai [8] 2021	38	N/A	CS	Intermittent abdominal pain, discomfort	18 months	Clinical *, MRI *	Soft mass about 5.5 × 4.4 cm^2^ in size in the left middle-lower abdomen	Yes	No
Bai [8] 2021	30	N/A	CS	Intermittent abdominal pain, abdominal mass	5 years	MRI *	Mass in the right middle and lower abdominal cavity 13.1 × 9.7 cm^2^	Yes	No
Munihire [52] 2022	31	I P	CS	Abdominal and pelvic pain, fever	22 days	USG *	7 cm diameter mesenteric mass	Yes	No
Ammar [53] 2021	36	III P	CS	Abdominal pain, vomiting	3 years	X-ray **	Retained sponge in the center of the abdomen	Resection and end-to-end anastomosis	Ileum perforation
Khanduri [54] 2022	38	I P	CS	Left iliac fossa pain, fever	1 month	USG *, CECT *	12 × 11 × 9 cm^3^ mass in left iliac fossa	Laparoscopy	No
Elci [55] 2021	29	II G, I P	Emergency CS	N/A	2 months	USG *	6.5 × 1.5 × 1 cm^3^ mass under the skin incision	Excision of the infected tissue	No
Min [56] 2022	54	II P	Emergency CS	Abdominal mass	19 years	USG **, CT *	10.4 cm pelvic mass, partially penetrated the right ovary	Yes	No
Our case 2021	28	I G, I P	Emergency CS supra-cervical HT—hematoma	Adnexal mass and subhepatic mass	16 months	USG *, MRI **	9.6 × 5.4 cm^2^ left flank mass, 8.5 × 4.9 cm^2^ right hypochondrium mass.	Yes	No

CS—cesarean section, HT—hysterectomy, USG—ultrasonography, CECT—contrast-enhanced computerized tomography, EGDS—esophago-gastroduodenoscopy, *—diagnosis, **—misdiagnosis.

**Table 2 jcm-12-05353-t002:** The diagnosis rate according to the type of imaging used compared to the studies that evaluated each technique.

Imagistic Findings	Diagnosis	Misdiagnosis
USG	18 (60%)	12 (40%)
X-ray	4 (25%)	12 (75%)
CT/CECT	27 (81.8%)	6 (18.2%)
MRI	6 (66.67%)	3 (33.33%)

**Table 3 jcm-12-05353-t003:** Complications of gossypiboma.

Complications	N	References
**Fistula**		
Ileocolic	3	[20,33,37]
Jejunocolic	2	[10,19]
Ileoileal	2	[14,32]
Descending colon	1	[27]
Jejunovaginal	1	[13]
Colouterine	1	[50]
Periumbilical	1	[23]
**Obstruction**		
Intestinal	3	[11,32,37]
**Perforations**		
Ileal	5	[1,26,29,51,53]
Jejunal	1	[45]
Sigmoid colon	1	[44]
**Bladder injury**	1	[34]

## Data Availability

Not applicable.

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
