# Peer review of "Gossypiboma, the Hidden Enemy of an Emergency Cesarean Hysterectomy—Case Report and Review of the Literature"

_jcm, 2023, doi:10.3390/jcm12165353_

Round 1
Reviewer 1 Report
Gossypiboma, the hidden enemy of an emergency caesarean hysterectomy- case report and literature review:
Summary- Gossypiboma is a serious but thankfully rare event. An important topic with significant morbidity and mortality. The medicolegal ramification is huge. Recognition of predisposing factors is important for prevention. There should be protocols and processes in place to ensure that instruments and sponges are accurate at the end of each surgery; thus, making this a “never event”.
The content of this manuscript is very relevant to all surgical team members especially in ObGyn where there is potential haemorrhage.
Whilst the manuscript is clear, References were balanced.
The methodology of the manuscript makes it easily reproducible. The conclusion highlights the need for policies to safeguard quality improvement and safety of patients.
I would like the authors to identify and report some limitations of this study.
.
Author Response
Gossypiboma, the hidden enemy of an emergency caesarean hysterectomy- case report and literature review:
Summary- Gossypiboma is a serious but thankfully rare event. An important topic with significant morbidity and mortality. The medicolegal ramification is huge. Recognition of predisposing factors is important for prevention. There should be protocols and processes in place to ensure that instruments and sponges are accurate at the end of each surgery; thus, making this a “never event”.
The content of this manuscript is very relevant to all surgical team members especially in ObGyn where there is potential haemorrhage.
Whilst the manuscript is clear, References were balanced.
The methodology of the manuscript makes it easily reproducible. The conclusion highlights the need for policies to safeguard quality improvement and safety of patients.
I would like the authors to identify and report some limitations of this study.
Answer: Thank you for appreciating our manuscript. We added the limitations section. We consider as limitations the fact that it is a subjective article that reports a rare pathology, with the risk of overinterpretation. (please see the attached manuscript) Lines 232-233.
Kindest regards
Reviewer 2 Report
This study showed an interesting case with a nice literature review. The reviewer thinks this case provided useful information for the readers. I have one suggestion to improve the methodology of this study. Please clarify the search date of the literature review (i.e. from the date of inception to May 31, 2023).
Almost good; however, some sentences are difficult to read and suggest to improve.
Author Response
This study showed an interesting case with a nice literature review. The reviewer thinks this case provided useful information for the readers. I have one suggestion to improve the methodology of this study. Please clarify the search date of the literature review (i.e. from the date of inception to May 31, 2023).
Answer: Thank you for appreciating our manuscript and for your remark. We are sorry that we omitted to mention the search data. We added it to the description section of the literature review (please see the attached manuscript), Lines 169-170.
Also, we corrected the quality of the English language of the manuscript.
Kindest regards
Authors